# Homogeneous Cosmological Models in Weyl's Geometrical Scalar–Tensor Theory

Adriano Barros [1] and Carlos Romero [2,*]

1 Centro de Desenvolvimento Sustentável do Semiárido, Universidade Federal de Campina Grande, Sumé 58540-000, PB, Brazil; atbarros@ufcg.edu.br
2 Departamento de Física, Universidade Federal da Paraíba, C.P. 5008, João Pessoa 58059-970, PB, Brazil
* Correspondence: cromero@fisica.ufpb.br

**Abstract:** In this paper, we consider homogeneous cosmological solutions in the context of the Weyl geometrical scalar–tensor theory. Firstly, we exhibit an anisotropic Kasner type solution taking advantage of some similarities between this theory and the Brans–Dicke theory. Next, we consider an isotropic model with a flat spatial section sourced by matter configurations described by a perfect fluid. In this model, we obtain an analytical solution for the stiff matter case. For other cases, we carry out a complete qualitative analysis theory to investigate the general behaviour of the solutions, presenting some possible scenarios. In this work, we do not consider the presence of the cosmological constant nor do we take any potential of the scalar field into account. Because of this, we do not find any solution describing the acceleration of the universe.

**Keywords:** Weyl geometry; scalar–tensor theory; cosmological models

## 1. Introduction

As is well known, scalar–tensor theories of gravity were proposed some years ago by Jordan [1], and Brans and Dicke [2]. Later, they were extended in a more general framework [3–5]. In fact, they represent a generalization of the simplest scalar–tensor theory of gravity which is the Brans–Dicke theory [6,7]. In general scalar–tensor theories of gravity, the gravitational field is not described only by the usual tensor field $g_{\mu\nu}$ of general relativity. In addition to this, we have one or several long-range scalar fields which also mediate gravitational interaction.

Scalar–tensor theories of gravity have been a subject of renewed interest. Certainly, one motivation for this is the belief that, at least at sufficiently high energy scales, gravity becomes scalar–tensorial in nature [8] and, therefore, these theories are important in the very early Universe. On the other hand, two important theoretical developments have been achieved such as, for example, unification models based on superstrings, which naturally associate long-range scalar partners to the usual tensor gravity of Einstein theory [9]. Another motivation for the investigation of scalar–tensor theories is that inflationary cosmology in this framework seems to solve the fine-tuning problem and, in this way, give us a mechanism of terminating inflationary eras [10]. Apart from the solution of this problem, the scalar–tensor theories by themselves have direct implications for cosmology and for experimental tests of the gravitational interaction [11] and for this reason, they are relevant in the investigation of the early Universe.

Among alternative theories of gravity, scalar–tensor theories are perhaps the most popular ones. As we have pointed out before, in these theories, gravitational effects are described by both a metric field $g_{\mu\nu}$ and a scalar field $\Phi$. A well-known example is the Brans–Dicke theory [2,12], in which the geometry of the underlying space-time manifold is assumed to be Riemannian, and the scalar field replaces the gravitational constant being interpreted as the inverse of a varying gravitational coupling parameter. In addition to the reasons mentioned above, the scalar–tensor theories are studied because they admit

key ingredients of string theories, such as a dilaton-like gravitational scalar field that has a non-minimal coupling to the curvature [13]. On the other hand, a different approach, in which the scalar field appears as part of the space-time geometry, namely, the Weyl geometrical scalar–tensor theory, has been discussed recently in the literature [14]. Indeed, in this new approach, one considers the space-time structure as a very special case of the framework adopted in the original Weyl unified field theory [15,16], the geometrical space-time structure being that of a *Weyl integrable space-time* (WIST) [17–21]. It is important to remark that other gravity theories in which a scalar field plays a geometrical role have also been proposed [22–24].

Recently, some theoretical aspects concerning the Weyl geometrical scalar–tensor theory have been studied, in particular the behaviour of the solutions when $\omega$, the scalar field's coupling constant, goes to infinity [25]. The investigation of cosmological vacuum models for different scalar potentials has also been carried out [26]. In the present article, we extend this research to include anisotropic models of Kasner type. Here, we take advantage of some similarities between vacuum solutions of the Weyl geometrical scalar–tensor theory and those coming from the Brans–Dicke theory. We also examine cosmological solutions in the presence of matter, a scenario that has not yet been investigated in Weyl geometrical scalar–tensor theory, and at the same time, we compare the results obtained with similar solutions already known from general relativity and the Brans–Dicke theory.

The paper is organized as follows. In Section 2, we briefly review Weyl's original theory, which inspired the geometrical scalar–tensor approach. In Section 3, the field equations of the Weyl geometrical scalar–tensor theory are obtained. Then, a Kasner type solution is exhibited in Section 4, while in Section 5 we work with a homogeneous and isotropic cosmological model having a perfect fluid as a source, such that we find an analytical solution for the stiff matter case and we study the other cases using the qualitative analysis of dynamical systems. Finally, Section 6 is devoted to our conclusions.

## 2. Weyl's Theory

In the first scalar–tensor theories, the so-called Jordan–Brans–Dicke theories, it is assumed, as in general relativity, that the space-time geometry is purely Riemannian. On the other hand, if the Palatini variational method is applied to deduce the field equations from the action, then in a large class of scalar–tensor theories, a non-Riemannian compatibility condition between the metric and the affine connection appears naturally (for a more general result, see [27]). In this way, we have a theory that establishes the space-time geometry from first principles, that is, the space-time manifold is dynamically generated by the choice of the particular coupling of the scalar field in the gravitational sector. In the case where the action is that of the Brans–Dicke theory, this procedure leads to the so-called *Weyl integrable space-times*, a particular version of the geometry conceived by H. Weyl in his attempt to unify gravity and electromagnetism [15]. Note, however, that here, it is the scalar field that is being geometrized.

It is true that the Weyl geometry is one of the simplest generalizations of Riemannian geometry, in which the Riemannian compatibility condition between the metric and the affine connection is weakened. This was an ingenious way that Weyl devised to introduce a covariant vector field $\sigma_\mu$ in the geometry, which bears a great similarity with the electromagnetic four-potential. Weyl went on and introduced the second-order tensor $F_{\mu\nu} = \partial_\mu \sigma_\nu - \partial_\nu \sigma_\mu$, which he interpreted as representing another kind of curvature, namely, the *length curvature*. As a consequence of this modification in the Riemannian compatibility condition, the covariant derivative of the metric tensor does not vanish, as in Riemannian geometry, and the length of vectors when parallel transported along a curve may change. However, such theory suffered from a severe criticism by Einstein, who objected that the nonintegrability of length implies that the rate at which a clock measures time, i.e., its clock rate, in this case would depend on the past history of the clock. As a consequence of this fact, spectral lines with sharp frequencies would not appear and the spectrum of neighbouring elements of the same kind would be different [28]. This became known

in the literature as the second clock effect (incidentally, the first clock effect refers to the well-known effect corresponding to the "twin paradox", which is predicted by special and general relativity theories).

Weyl's new compatibility condition is given by $\nabla_\alpha g_{\mu\nu} = \sigma_\alpha g_{\mu\nu}$, and is easily verified that this condition is invariant under the conformal transformation $g_{\mu\nu} \to \bar{g}_{\mu\nu} = e^f g_{\mu\nu}$ carried out simultaneously with the *gauge* transformation $\sigma_\mu \to \bar{\sigma}_\mu = \sigma_\mu + \partial_\mu f$, where $f$ is an arbitrary scalar function. The discovery of this new symmetry is now considered by some authors as the birth of modern gauge theories [29]. Now, if $F_{\mu\nu} = 0$ (null second curvature), which is equivalent to say that the one-form $\sigma$ is closed ($d\sigma = 0$), then there is no electromagnetic field. In this case, we know that, from Poincaré's lemma [30], it follows that there exists a scalar field $\phi$, such that $\sigma_\mu = \partial_\mu \phi$, and, instead of a vector field $\sigma$, we are left with a scalar field $\phi$, which, in addition to the metric, is the fundamental object that characterizes the geometry. A space-time endowed with this particular version of Weyl's geometry came to be known as a *Weyl integrable space-time*.

## 3. The Field Equations

As we have already mentioned, in the Weyl geometrical scalar–tensor theory, the underlying space-time manifold is that of a Weyl integrable space-time [17]. In this sense, the Weyl nonmetricity condition involves a purely geometrical scalar field $\phi$ and is explicitly given by [14]

$$\nabla_\alpha g_{\mu\nu} = g_{\mu\nu}\phi_{,\alpha}. \tag{1}$$

Moreover, one can define the Weyl connection, whose coefficients in a local coordinate basis read

$$\Gamma^\alpha_{\mu\nu} = \{^\alpha_{\mu\nu}\} - \frac{1}{2}g^{\alpha\beta}(g_{\beta\mu}\phi_{,\nu} + g_{\beta\nu}\phi_{,\mu} - g_{\mu\nu}\phi_{,\beta}), \tag{2}$$

with $\{^\alpha_{\mu\nu}\}$ representing the usual Christoffel symbols.

In turn, the field equations of the Weyl geometrical scalar–tensor theory can be written as [25]

$$G_{\mu\nu} = -\frac{(\omega - \frac{3}{2})}{\Phi^2}\left(\Phi_{,\mu}\Phi_{,\nu} - \frac{g_{\mu\nu}}{2}\Phi_{,\alpha}\Phi^{,\alpha}\right)$$
$$- \frac{1}{\Phi}(\Phi_{,\mu;\nu} - g_{\mu\nu}\Box\Phi) - \frac{g_{\mu\nu}}{2\Phi}V(\Phi) - 8\pi T_{\mu\nu}, \tag{3}$$

$$\Box\Phi = \frac{1}{\omega}\left(-\frac{1}{2}\frac{dV}{d\Phi}\Phi + V(\Phi)\right), \tag{4}$$

where here, we are using the field variable $\Phi = e^{-\phi}$, $\omega = const$, $V(\phi)$ corresponds to the scalar field potential, and $T_{\mu\nu}$ represents the Weyl invariant energy–momentum tensor of the matter fields. We denote by $G_{\mu\nu}$ and $\Box$ the Einstein tensor and the d'Alembertian operator, respectively, defined with respect to the Christoffel symbols. If $V(\Phi) = 2\Lambda\Phi$, one can introduce the cosmological constant $\Lambda$. However, let us take $\Lambda = 0$, and then the field equations are given by

$$G_{\mu\nu} = -\frac{W}{\Phi^2}\left(\Phi_{,\mu}\Phi_{,\nu} - \frac{1}{2}g_{\mu\nu}\Phi_{,\alpha}\Phi^{,\alpha}\right)$$
$$- \frac{1}{\Phi}\Phi_{,\mu;\nu} - 8\pi T_{\mu\nu}, \tag{5}$$

$$\Box\Phi = 0, \tag{6}$$

where $W = \omega - \frac{3}{2}$. Additionally, we can obtain from (5) and (6) that

$$R_{\mu\nu} = -8\pi T_{\mu\nu} + \frac{8\pi T}{2}g_{\mu\nu} - \frac{W}{\Phi^2}\Phi_{,\mu}\Phi_{,\nu} - \frac{\Phi_{,\mu;\nu}}{\Phi}, \tag{7}$$

with $R_{\mu\nu}$ denoting the Ricci tensor and $T = g_{\mu\nu}T^{\mu\nu}$. Equations (6) and (7) constitute the field equations we use in the following.

## 4. Kasner Type Solution

As is well known, the Kasner metric was obtained by the mathematician E. Kasner in 1921 and represents an exact solution to Einstein's field equations. It describes an anisotropic universe without matter, that is, it is a vacuum solution. Historically, interest in the Kasner solution came from the fact that, although it may have a singularity ("big bang" or a "big crunch"), an isotropic expansion or contraction of space is not allowed, and this led to the generic singularity studies, the so-called BKL singularities [31].

The Kasner type solution in the Brans–Dicke theory of gravity is given by [2,32]

$$ds^2 = dt^2 + R_1^2 dx^2 + R_2^2 dy^2 + R_3^2 dz^2, \tag{8}$$

with

$$R_i = r_i(at+b)^{\frac{p_i}{1+C}}, \tag{9}$$

$(i = 1, 2, 3)$ and the Brans–Dicke scalar field

$$\varphi = \varphi_0(at+b)^{\frac{C}{1+C}}, \tag{10}$$

where $a$, $b$, $r_i$, and $\varphi_0$ are constants. The relations $\sum p_i = 1$ and

$$\sum p_i^2 = 1 - C(\omega C - 2) \tag{11}$$

between the constants $p_i$, $C$ and the scalar field coupling constant $\omega$ are also satisfied.

The space-time given by (8) corresponds to a homogeneous universe, without matter and rotation, with distinct expansions along the three orthogonal axes, which reflects anisotropy. Note that if $a = 1$ and $b = 0$, Equations (9) and (10) may be written as

$$R_i = r_i t^{\frac{p_i}{1+C}}, \tag{12}$$

$$\varphi = \varphi_0 t^{\frac{C}{1+C}}. \tag{13}$$

In order to obtain a solution in the Weyl geometrical scalar–tensor theory, let us consider the following result: a vacuum solution of the Weyl geometrical scalar–tensor theory can be found if we make the change $\omega \rightarrow W = \omega - 3/2$ in the correspondent vacuum solution of the Brans–Dicke theory. In fact, the two theories are not physically equivalent given that in Weyl's geometrical scalar–tensor theory test particles follow affine Weyl geodesics (autoparallels) and not metric geodesics as in the case of the Brans–Dicke theory. Nonetheless, there is a formal equality between the vacuum field equations of the two theories [14].

Thus, the Kasner type solution in the Weyl geometrical scalar–tensor theory is given by Equation (12) and

$$\Phi = \Phi_0 t^{\frac{C}{1+C}}, \tag{14}$$

where $\sum p_i = 1$ and

$$\sum p_i^2 = 1 - C(WC - 2) = 1 - C\left[\left(\omega - \frac{3}{2}\right)C - 2\right]. \tag{15}$$

Now, if we choose $C = \dfrac{2}{W}$, it follows that

$$\sum p_i^2 = 1. \tag{16}$$

Furthermore, (12) and (14) become

$$R_i = r_i t^{\frac{W p_i}{W+2}} = r_i t^{[(\omega-3/2)/(\omega+1/2)]p_i},\tag{17}$$

$$\Phi = \Phi_0 t^{\frac{2}{W+2}} = \Phi_0 t^{[2/(\omega+1/2)]}.\tag{18}$$

In the limit $\omega \to \infty$, (17) and (18) tend to

$$R_i = t^{p_i},\tag{19}$$

$$\Phi = \Phi_0,\tag{20}$$

where we have taken $r_i = 1$. On the other hand, from (1) and (2) we find that

$$\nabla_\alpha g_{\mu\nu} = -g_{\mu\nu}\left(\frac{\Phi_{,\alpha}}{\Phi}\right),\tag{21}$$

$$\Gamma^\alpha_{\mu\nu} = \{^{\ \alpha}_{\mu\nu}\} + \frac{1}{2\Phi}g^{\alpha\beta}\left(g_{\beta\mu}\Phi_{,\nu} + g_{\beta\nu}\Phi_{,\mu} - g_{\mu\nu}\Phi_{,\beta}\right),\tag{22}$$

by considering the scalar field in the form $\Phi = e^{-\phi}$. Thus, when $\omega \to \infty$, the space-time geometry becomes Riemannian as we have

$$\nabla_\alpha g_{\mu\nu} = 0, \ \text{ and } \ \Gamma^\alpha_{\mu\nu} = \{^{\ \alpha}_{\mu\nu}\}.\tag{23}$$

Therefore, also taking into account (19) and (20), the Kasner solution of general relativity is recovered in this limit.

## 5. A Perfect Fluid Cosmological Model

The Friedmann–Robertson–Walker metric with a flat spatial section is given by

$$ds^2 = dt^2 - R^2(t)\left[dr^2 + r^2\left(d\vartheta^2 + \sin^2\vartheta d\chi^2\right)\right],\tag{24}$$

where $R(t)$ denotes the scale factor. In this cosmological model, the matter content is a perfect fluid represented by the energy–momentum tensor

$$T_{\mu\nu} = (p + \rho)u_\mu u_\nu - p g_{\mu\nu},\tag{25}$$

with $p = \lambda\rho$, $0 \leq \lambda \leq 1$, $p$ being the thermodynamic pressure, $\rho$ the energy density, and $u_\mu = (1, 0, 0, 0)$ the four-velocity vector field. Then, field Equations (6) and (7) reduce to

$$\frac{3\ddot{R}}{R} = -4\pi\rho(1 + 3\lambda) - W\frac{\dot\Phi^2}{\Phi^2} - \frac{\ddot\Phi}{\Phi},\tag{26}$$

$$\frac{\ddot{R}}{R} + \frac{2\dot{R}^2}{R} = 4\pi\rho(1 - \lambda) - \frac{\dot{R}\dot\Phi}{R\Phi},\tag{27}$$

$$\frac{\ddot\Phi}{\Phi} + \frac{3\dot{R}\dot\Phi}{R\Phi} = 0.\tag{28}$$

The dot means differentiation with respect to time. Moreover, due to the assumption of spatial homogeneity, the scalar field $\Phi$ is supposed to be a function of $t$ only. Additionally, with the definitions $\theta = \frac{3\dot{R}}{R}$ and $\Psi = \frac{\dot\Phi}{\Phi}$, one can express (26)–(28) in the form

$$\dot\theta = -\frac{\theta^2}{3} - 4\pi\rho(1 + 3\lambda) - (W + 1)\Psi^2 - \dot\Psi,\tag{29}$$

$$\dot{\theta} = -\theta^2 + 12\pi\rho(1-\lambda) - \theta\Psi, \tag{30}$$

$$\dot{\Psi} = -\Psi^2 - \theta\Psi. \tag{31}$$

By combining (29)–(31), we can derive the equation

$$\frac{\theta^2}{3} - \frac{W\Psi^2}{2} + \theta\Psi = 8\pi\rho. \tag{32}$$

After some calculations and by using Equations (5) and (6), it is easy to show that

$$T^{\mu\nu}{}_{;\nu} = \frac{T}{2}\frac{\Phi^{,\mu}}{\Phi} - \frac{\Phi_{,\nu}}{\Phi}T^{\mu\nu}, \tag{33}$$

which reduces to

$$\dot{\rho} = -\left[(1+\lambda)\theta + \left(\frac{1+3\lambda}{2}\right)\Psi\right]\rho \tag{34}$$

in the context of the cosmological model considered.

### 5.1. Stiff Matter Solution

Next, we obtain the equations of a dynamic system which lead us to carry out a rich analysis of the solutions. For this purpose, let us consider the following equation, which results from (29)–(31):

$$\dot{\theta} = -\frac{(1+\lambda)}{2}\theta^2 + \frac{(1-3\lambda)}{2}\theta\Psi - \frac{3W(1-\lambda)}{4}\Psi^2. \tag{35}$$

This equation, together with (31), constitutes a homogeneous autonomous planar dynamic system. It is important to note that the solutions of this system, $\theta(t)$ and $\Psi(t)$, must necessarily satisfy the constraint imposed by Equation (32).

Cosmological scenarios modelled by stiff matter have been investigated recently, particularly in connection with the problem of dark matter [33]. Now, let us consider the stiff matter case in the geometrical scalar–tensor theory. Then, it follows from (35), in the case known as stiff matter ($\lambda = 1$), that

$$\dot{\theta} = -\theta^2 - \theta\Psi. \tag{36}$$

Clearly, an immediate solution of the system of Equations (31) and (36) is given by $\Psi = -\theta$, which leads to the particular solution ($\theta = \theta_0$, $\Psi = \Psi_0$), $\theta_0$ and $\Psi_0$ being constants. Hence, we have $\frac{3\dot{R}}{R} = \theta_0$, $\frac{\dot{\Phi}}{\Phi} = \Psi_0$, which then leads to

$$R(t) = R_0 \exp\left(\frac{\theta_0 t}{3}\right), \tag{37}$$

$$\Phi(t) = \Phi_0 \exp(\Psi_0 t), \tag{38}$$

where $R_0$ and $\Phi_0$ are constants, which we recognize as a de Sitter type solution, with the scalar field also having an exponential behaviour. Furthermore, from (32), we can find

$$\rho = -\frac{(3W+4)}{48\pi}\theta_0^2. \tag{39}$$

Now, by defining $\alpha = \theta + \Psi \neq 0$, let us find the general solution for the stiff matter case. To do this, one can add Equations (31) and (36) to obtain

$$\dot{\alpha} + \alpha^2 = 0, \tag{40}$$

whose solution is

$$\alpha = \frac{1}{t + D},$$ (41)

where $D$ is a constant. In turn, from Equation (34) with $\lambda = 1$, it follows that

$$\dot{\rho} + 2\alpha\rho = 0,$$ (42)

whose solution is

$$\rho = \frac{\rho_0}{(t + D)^2},$$ (43)

with $\rho_0$ constant.

To obtain the expression of $\theta$, let us consider (32) and (43) and use that $\Psi = \alpha - \theta = \frac{1}{t+D} - \theta$. In this way we are led to

$$\theta = \frac{B}{t + D},$$ (44)

$$\Psi = \frac{1 - B}{t + D},$$ (45)

where

$$B = \frac{3(W + 1) \pm \sqrt{3[(3 + 2W) - 16\pi\rho_0(4 + 3W)]}}{4 + 3W},$$ (46)

while the condition $(3 + 2W) - 16\pi\rho_0(4 + 3W) \geq 0$ is required to be satisfied.

On the other hand, if we replace (44) and (45) in (32), we obtain

$$\rho = \frac{1}{8\pi(t + D)^2}\left[\frac{B^2}{3} - \frac{W}{2}(1 - B)^2 + B(1 - B)\right].$$ (47)

The solutions for the scale factor and the scalar field can be obtained by integrating the expressions $\theta = \frac{3\dot{R}}{R}$ and $\Psi = \frac{\dot{\Phi}}{\Phi}$, giving the following:

$$R(t) = R_0(t + D)^{B/3},$$ (48)

$$\Phi(t) = \Phi_0(t + D)^{1-B},$$ (49)

with $R_0$ and $\Phi_0$ being constants.

It should be noted that the constant $B$ can also be written as

$$B = 1 - \frac{1}{4 + 3W} \pm \frac{\sqrt{3[(3 + 2W) - 16\pi\rho_0(4 + 3W)]}}{4 + 3W}.$$ (50)

Thus, for a large $W$, we obtain

$$B = 1 - \frac{1}{3W} \pm \sqrt{\frac{1}{3W}(2 - 48\pi\rho_0)}.$$ (51)

Let us now consider that $\rho_0 = \frac{f(W)}{24\pi}$, $f(W)$ being a function which tends to one when $W$ is large. Therefore, $B$ takes the form

$$B = 1 - \frac{1}{3W} = 1 + O\left(\frac{1}{W}\right).$$ (52)

Under these conditions, one can obtain from (49) that

$$\Phi(t) = \Phi_0 + O\left(\frac{1}{W}\right).$$ (53)

When $\Phi$ behaves as in (53) for a large $W$, it has been verified that any vacuum solution of the Weyl geometrical scalar–tensor theory reduces to the corresponding general relativistic solution in the limit $W \to \infty$ [25]. This fact also occurs here, since Equations (48) and (49) become equal to the Einstein solution

$$R(t) = R_0 t^{1/3}, \tag{54}$$

$$\Phi = \Phi_0, \tag{55}$$

for $W \to \infty$ (we take $D = 0$). Naturally, the geometry of the space-time becomes Riemannian, according to Equations (21)–(23).

*5.2. Qualitative Analysis for $\lambda \neq 1$*

For values of the parameter $\lambda$ in the interval $0 \leq \lambda < 1$, we use the qualitative analysis theory [34], by which many of the general characteristics of the integral solutions of the system can be studied without working out explicit solutions $\theta(t)$ and $\Psi(t)$. For this purpose, let us start by writing Equations (31) and (35) of the dynamic system as

$$\dot{\theta} = F(\theta, \Psi) = -\frac{(1+\lambda)}{2}\theta^2 + \frac{(1-3\lambda)}{2}\theta\Psi - \frac{3W(1-\lambda)}{4}\Psi^2, \tag{56}$$

$$\dot{\Psi} = H(\theta, \Psi) = -\Psi^2 - \theta\Psi. \tag{57}$$

An equilibrium point of the system, i.e., a solution that occurs when $F(\theta, \Psi) = H(\theta, \Psi) = 0$, is the origin of the phase plane, the point $M$ ($\theta = 0, \Psi = 0$). This solution represents Minkowski's space-time, being the only finite equilibrium point that is significant in the system.

In the qualitative analysis of solutions of Equations (56) and (57), one must construct the phase diagrams. For this, we make use of the Poincaré compactification method, which projects the phase plane into a sphere. A second mapping, in turn, projects this sphere orthogonally onto a disk, whose circumference represents the infinity of the initial phase plane [34].

*5.3. Invariant Rays and Regions of Negative Energy Density*

Initially, in our analysis, we obtain the invariant rays of the dynamic system defined above. For this, let us make the change of variables $\theta = r \cos \beta$ and $\Psi = r \sin \beta$, $r$ and $\beta$ being polar coordinates of the plane. In this way, we find

$$\dot{\theta} = r^2 \left[ -\frac{(1+\lambda)}{2} \cos^2 \beta + \frac{(1-3\lambda)}{2} \cos \beta \sin \beta - \frac{3W(1-\lambda)}{4} \sin^2 \beta \right] = r^2 \overline{F}(\beta), \tag{58}$$

$$\dot{\Psi} = r^2 \left[ -\sin^2 \beta - \cos \beta \sin \beta \right] = r^2 \overline{H}(\beta). \tag{59}$$

Now, from the relations between the variables $\theta$, $\Psi$, $r$, $\beta$, and Equations (58) and (59), it can be shown that

$$\dot{\beta} = r \left( -\overline{F}(\beta) \sin \beta + \overline{H}(\beta) \cos \beta \right). \tag{60}$$

Next, we obtain the invariant rays, which, by definition, consist of solutions where the ratio $\frac{\Psi}{\theta} = \tan \beta = const$. Thus, putting $\dot{\beta} = 0$ in expression (60) leads to

$$\tan \beta = \frac{\overline{H}(\beta)}{\overline{F}(\beta)}. \tag{61}$$

Again, with the help of Equations (58) and (59), it follows from (61) that

$$\tan\beta\left(\frac{W}{2}\tan^2\beta - \tan\beta - \frac{1}{3}\right) = 0. \tag{62}$$

For $W < -\frac{3}{2}$, the roots of (62) are $\beta_1 = 0$ and $\beta_2 = \pi$. The solutions representing these invariant rays appear in phase diagrams such as curves $AM$ and $MA'$, respectively (see Figure 1, for example). When $W > -\frac{3}{2}$, in addition to the roots $\beta_1$ and $\beta_2$ already mentioned, there are four more:

$$\beta_3 = \tan^{-1}\left[-\frac{3}{2}\left(1 + \sqrt{1 + \frac{2W}{3}}\right)\right]^{-1}, \quad \beta_4 = \beta_3 + \pi, \tag{63}$$

$$\beta_5 = \tan^{-1}\left[-\frac{3}{2}\left(1 - \sqrt{1 + \frac{2W}{3}}\right)\right]^{-1}, \quad \beta_6 = \beta_5 + \pi, \tag{64}$$

which correspond to the curves $BM$, $MB'$, $CM$, and $MC'$, respectively (see Figure 2, for instance). These invariant rays depend on $W$ and, as its value increases, the following behaviour is observed: the line $BB'$ rotates anticlockwise approaching the $\theta$-axis, while the line $CC'$ moves clockwise tending to make an angle of $-180°$ with the positive direction of the $\theta$-axis. It should also be noted that if $W = -\frac{3}{2}$, the lines $BB'$ and $CC'$ coincide, making an angle of $-33.69°$ with the $\theta$-axis.

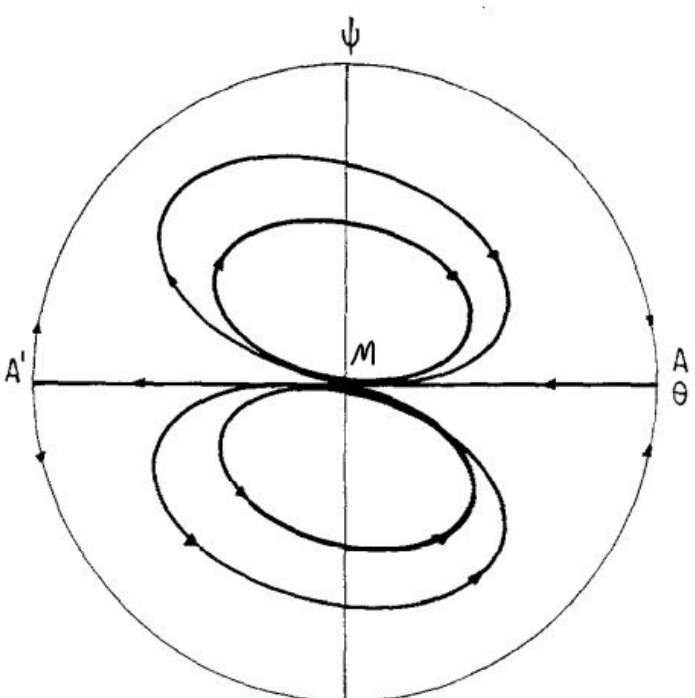

**Figure 1.** $W < -\frac{3}{2}$ $(\omega < 0)$.

To continue, let us check if there are regions of the phase diagrams in which $\rho < 0$. In these regions, the solutions should not be admitted as physical solutions, at least classically.

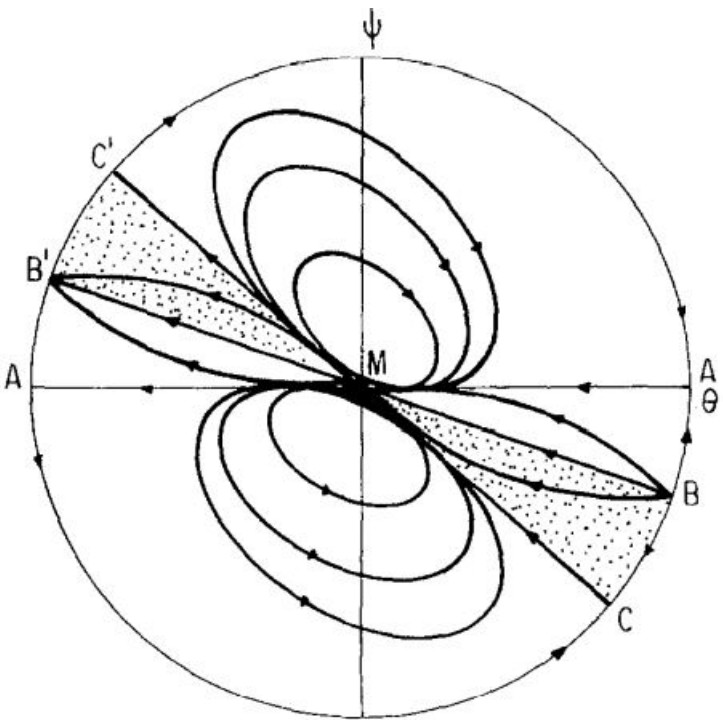

**Figure 2.** $-\frac{3}{2} < W < -\frac{4}{3}$ ($0 < \omega < \frac{1}{6}$).

We start by replacing $\Psi = \theta \tan \beta$ in (32). We thus obtain

$$-\theta^2 \left( \frac{W}{2} \tan^2 \beta - \tan \beta - \frac{1}{3} \right) = 8\pi\rho. \tag{65}$$

It is easy to verify, taking into account (62), that the invariant rays lying on the lines $BB'$ and $CC'$ represent vacuum solutions. Moreover, we have no region with a negative energy density if $W < -\frac{3}{2}$. On the other hand, when $W > -\frac{3}{2}$, we find regions where $\rho < 0$ that are delimited by the invariant rays that lie on the lines $BB'$ and $CC'$. In the next section, these regions are represented as dotted regions in the phase diagrams, which widen as the value of $W$ increases, tending to leave the classically allowed solutions localized in a narrow region that includes the $\theta$-axis.

*5.4. Phase Diagrams*

Now, one can obtain the basic representation of Weyl's cosmological solutions on the Poincaré sphere (the phase diagrams). This allows us to make a qualitative analysis of the solutions at infinity. First, let us make some comments about the diagrams (Figures 1–3), which are valid for $\lambda \neq 1$ and are separated into intervals of $W$ (or $\omega$)[1].

Initially, for $W < -3/2$ (see Figure 1), the closed curves appearing in the diagram represent nonsingular cosmological models, which start in the infinitely distant past from Minkowski's space-time (the point $M\,(0,0)$) and tend to it again in the infinitely distant future; these universes present an initial phase of contraction, and then move into an expansive phase. For some of these solutions, the scalar field $\Phi$ is increasing (if $\Psi > 0$), while for the others, it is decreasing, in which case $\Psi < 0$. On the other hand, it is possible to have singular solutions with a constant scalar field ($\Psi = 0$): they are represented by the $AM$ curves, which correspond to solutions that start with a "big bang", and then undergo an expansive phase, finally tending to Minkowski's space-time, and the $MA'$ curves, solutions that start from Minkowski's space-time (in the infinitely distant past, with the cosmic time $t \to -\infty$), and follow a contraction regime until the final collapse.

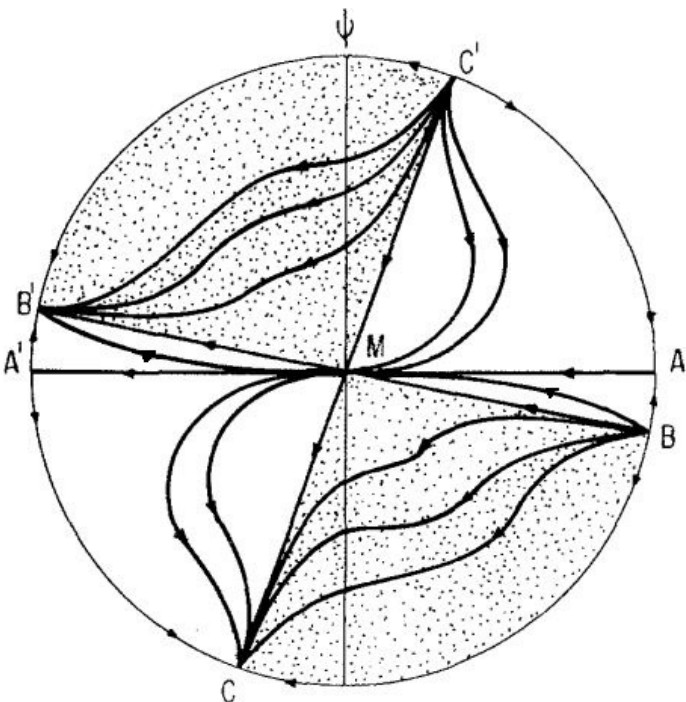

**Figure 3.** $W > 0$ $(\omega > \frac{3}{2})$.

In fact, the curves $AM$ and $MA'$ also correspond to solutions of general relativity, since from (56) with $\Psi = 0$, it follows that

$$\dot{\theta} = -\frac{(1+\lambda)}{2}\theta^2, \tag{66}$$

whose solution is

$$\frac{1}{\theta} = \frac{(1+\lambda)}{2}t + \delta, \tag{67}$$

where $\delta$ is an arbitrary constant. Therefore, by setting $\delta = 0$, we obtain the known scale factor

$$R(t) = R_0 t^{2/3(1+\lambda)}. \tag{68}$$

In Figure 2, we consider the interval $-3/2 < W < -4/3$. In this diagram, there are six invariant rays: $AM$, $MA'$, $BM$, $MB'$, $CM$, and $MC'$. It is interesting to recall that the dotted regions in the diagram contain solutions with $\rho < 0$, so that the curves restricted to these regions do not correspond to physical models. Furthermore, solutions lying on the lines $BB'$ and $CC'$ are vacuum solutions ($\rho = 0$), possessing singularities in their geometries, i.e., they are "big bang" models ($BM$ and $CM$) or models that collapse ($MB'$ and $MC'$), but with the scalar field varying. In the region where $\rho > 0$, one finds solutions similar to those in the previous diagram and also expanding universes with decreasing $\Phi$ ($BM$) and collapsing universes with increasing $\Phi$ ($MB'$).

For $W > -\frac{4}{3}$, it turns out that there are no nonsingular solutions in the diagrams. In Figure 3 ($W > 0$), in addition to solutions that appeared in Figure 2 when $\rho \geq 0$, we now observe the existence of expanding universes with increasing $\Phi$ ($C'M$) and collapsing universes with decreasing $\Phi$ ($MC$). As mentioned before, if $W$ increases, the line $BB'$ moves anticlockwise approaching the line $AA'$, while the line $CC'$ moves clockwise, also approaching $AA'$; as a consequence, the "forbidden" regions (sectors $MB'C'$ and $MBC$), where $\rho < 0$, become wider. In the limit $W \to \infty$, the line $AA'$ remains in the region where the energy density $\rho$ is positive, representing the solutions of general relativity given by (68). Actually, for each value of $W$, the line $AA'$ contains the solutions (68) because $\Phi = const$ (which implies $\Psi = 0$) is a solution to Equation (6).

In most of the diagrams, the equilibrium points do not appear as isolated points. In these cases, they correspond to multiple equilibrium points, constituting the invariant rays. In the other cases, they appear on the Poincaré sphere as points at the infinity, whose nature are indicating in Table 1 below.

**Table 1.** Behaviour of the equilibrium points on the Poincaré sphere.

| Intervals | $A, A'$ | $B, B'$ | $C, C'$ |
|---|---|---|---|
| $W < -3/2$ | Saddle points | - | - |
| $-3/2 < W < -4/3$ | Saddle points | Two-tangent nodes | Saddle points |
| $W > -4/3$ ($W \neq 0$) | Saddle points | Two-tangent nodes | Two-tangent nodes |

## 6. Conclusions

In this paper, we sought to find cosmological solutions in the context of the Weyl geometrical scalar–tensor theory. The vacuum field equations of this theory are formally identical to those of the Brans–Dicke theory, so we were able to obtain a Kasner type solution from the corresponding solution in the Brans–Dicke theory. We also found that, in the limit $\omega \to \infty$, the Kasner solution of general relativity was recovered. On the other hand, we investigated the existence of solutions for homogeneous and isotropic models sourced by a perfect fluid. In this case, we found an analytic solution for stiff matter and also showed that the corresponding solution of general relativity could be obtained in the limit $W \to \infty$. For values of the parameter $\lambda \neq 1$, no analytical solution was possible, and we used dynamical systems theory to display the phase diagrams of the solutions in intervals of $W$ (or $\omega$). When $W > 0$, we highlighted solutions representing universes with $\rho > 0$ and an increasing geometric scalar field, which started with a "big bang" and expanded to a final phase that tended toward Minkowski's space-time (the curves $C'M$).

An interesting fact regarding the phase diagrams examined here is that there was no difference between the cosmological models when different values of the parameter $\lambda$ were considered. In that sense, it can be seen that Equation (62), which determines the invariant rays, did not depend on $\lambda$. Moreover, it should be noted that in the present context, matter was not a source of the geometric scalar field $\Phi$ in Equation (6). By contrast, in the Brans–Dicke theory, the scalar field equation is

$$\Box \varphi = \frac{8\pi T}{2\omega + 3},$$ (69)

where, as is well known, $T$ denotes the trace of the energy–momentum tensor. For the case of a perfect fluid source, $T = T(\lambda)$ and $T = 0$ only when $\lambda = \frac{1}{3}$. As a consequence, in the present scalar–tensor theory, cosmological models differed according to the value of the parameter $\lambda$ [35].

In this work, we did not consider the presence of the cosmological constant, nor did we take any potential of the scalar field into account. Because of this, we did not find any solution describing the acceleration of the universe. Incidentally, models describing cosmological scenarios in which the acceleration of the cosmos is driven by a scalar field, quintessence models [36,37] and Chaplygin gas models [38,39] among others [40], have been investigated with interest. Two lines of research that we leave for further work are (i) an investigation of the role the geometric scalar field could play in approaching the problem of dark matter and (ii) considering scenarios where the cosmological constant is present with the hope that they can give some light to the problem of dark energy.

**Author Contributions:** The authors contributed equally to this work. All authors have read and agreed to the published version of the manuscript.

**Funding:** C. Romero is partially supported by Conselho Nacional de Desenvolvimento Científico e Tecnológico (CNPq-Brazil) through the Research Project No. 310046/2022-0.

**Data Availability Statement:** Not applicable.

**Acknowledgments:** We would like to express our gratitude to the referees for their valuable comments and suggestions.

**Conflicts of Interest:** The authors declare no conflict of interest.

## Note

1　　The cases $W = -\frac{3}{2}$, $W - \frac{4}{3}$ and $W = 0$ were not analysed because they contain multiple equilibrium points or singularities.

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
