# Peer review of "Homogeneous Cosmological Models in Weyl’s Geometrical Scalar–Tensor Theory"

_universe, doi:10.3390/universe9060283_

Round 1

Reviewer 1 Report

The Weyl scalar-tensor theory is quite interesting extension of general relativity, which is worth to be studied in details. The authors consider homogeneous cosmological solutions in the context of the Weyl geometrical scalar-tensor theory and obtain some new results. Namely, they found a vacuum anisotropic Kasner-type solution, which turned out similar to that in Brans-Dicke theory. As well they considered isotropic cosmological models with flat spatial section sourced by a perfect fluid. I found the paper to be quite interesting, well written, consisting new physical results, and so I can recommend the paper for publication in Universe.

Author Response

Thank you for your comments

Reviewer 2 Report

In the draft for review, the authors have studied homogeneous cosmology in Weyl Scalar-Tensor theory. First, they obtained an exact analytic solution assuming stiff matter, and after that they investigated the general behavior of the model including phase-space dynamics techniques. The manuscript is well written and clearly presented, although I find the bibliography too short. Moreover, despite the fact that the phase-space diagrams are shown, I have not seen a table summarizing the critical points and their nature (attractor, unstable, saddle) for each case.

Before I recommend the submitted  manuscript for publication, I would like to ask the authors to i) extend the bibliography considering the richness of the topic, ii) add the tables showing the critical points and their nature for each case, and iii) commenting on whether or not there is a stable point corresponding to an accelerating Universe. This last piece of information should be included in the conclusions as well as in the abstract.

Therefore, I am asking for minor revisions.

Author Response

Following the suggestion by the referee,

a)     We have substantially increased the introduction and added many news references;

b)    We have added a table describing all the critical points which appear in the phase diagrams;

c)     We have answered the question whether there is no stable equilibrium point corresponding to a solution describing an accelerated universe. We also have commented on the reasons why we would not expect to find a solution corresponding to an accelerated cosmos. These comments are included both in the abstract and in the conclusions.

d)    We have added a new section on the Weyl theory for clarity reasons, which we hope will help the reading of the article.

e)     We would like to thank the referee for his/her relevant and useful comments. 

Reviewer 3 Report

In this manuscript, the authors have obtained homogeneous cosmological solutions in the context of the Weyl geometrical scalar-tensor theory. The similarities between this theory and Brans-Dicke theory are taken into account in obtaining an anisotropic Kasner-type solution. An isotropic matter with a perfect fluid is also considered, and an analytical solution for the case is also obtained. The general behaviours of obtained solutions are investigated.

I have a comment/question on the relevance of the solution. I think using the solutions one may explain expansion of the Universe and Hubble constant describing the nature of dark energy. 

Author Response

Following the suggestion by the referee,                        

a) We have increased the introduction, the abstract, the conclusions, and added news references.                       

b) We have commented on the fact that there is no stable equilibrium point corresponding to a solution describing an accelerated universe  and why we would not expect to find a solution corresponding to such an accelerated cosmos. Therefore, the solutions found are not appropriate for describing the dark energy era of the universe. As we say in the text, this is due to the fact that the cosmological constant is not included in the field equations. We have the investigation of a more complete scenario, which would include a cosmological constant or/and quintessence scalar potential for future research.

 c) We have added a new section on the Weyl theory for clarity reasons, which we hope will help the reading of the article.

d) We would like to thank the referee for his/her relevant and useful comments.